# Carbide Precipitation during Processing of Two Low-Alloyed Martensitic Tool Steels with 0.11 and 0.17 V/Mo Ratios Studied by Neutron Scattering, Electron Microscopy and Atom Probe

Erik Claesson [1,2,*], Hans Magnusson [2], Joachim Kohlbrecher [3], Mattias Thuvander [4], Fredrik Lindberg [2], Magnus Andersson [2] and Peter Hedström [1]

[1] Department Materials Science and Engineering, KTH Royal Institute of Technology, SE-10044 Stockholm, Sweden; pheds@kth.se

[2] Swerim AB, Box 7047, Kista, SE-16407 Stockholm, Sweden; hans.magnusson@swerim.se (H.M.); fredrik.lindberg@swerim.se (F.L.); magnus.andersson@swerim.se (M.A.)

[3] Laboratory of Neutron Scattering and Imaging, Paul Scherrer Institute, 5232 Villigen, Switzerland; joachim.kohlbrecher@psi.ch

[4] Department of Physics, Chalmers University of Technology, SE-41296 Gothenburg, Sweden; mattias.thuvander@chalmers.se

[*] Correspondence: erik.claesson@swerim.se

**Abstract:** Two industrially processed low-alloyed martensitic tool steel alloys with compositions Fe-0.3C-1.1Si-0.81Mn-1.5Cr-1.4Ni-1.1Mo-0.13V and Fe-0.3C-1.1Si-0.81Mn-1.4Cr-0.7Ni-0.8Mo-0.14V (wt.%) were characterized using small-angle neutron scattering (SANS), scanning electron microscopy (SEM), Scanning transmission electron microscopy (STEM), and atom probe tomography (APT). The combination of methods enables an understanding of the complex precipitation sequences that occur in these materials during the processing. Nb-rich primary carbides form at hot working, while Fe-rich auto-tempering carbides precipitate upon quenching, and cementite carbides grow during tempering when Mo-rich secondary carbides also nucleate and grow. The number density of Mo-rich carbides increases with tempering time, and after 24 h, it is two to three orders of magnitude higher than the Fe-rich carbides. A high number density of Mo-rich carbides is important to strengthen these low-alloyed tool steels through precipitation hardening. The results indicate that the Mo-rich secondary carbide precipitates are initially of MC character, whilst later they start to appear as $M_2C$. This change of the secondary carbides is diffusion driven and is therefore mainly seen for longer tempering times at the higher tempering temperature of 600 °C.

**Keywords:** tool steel; precipitation; carbides; martensite; tempering; STEM; APT; SANS



## 1. Introduction

There is a continuous interest in improving the mechanical properties of high-performance steels grades used in tooling applications such as plastic extrusion. These materials rely on a high hardness originating from the precipitation of carbides and a martensitic structure. The industrial materials included in this study are produced in a continuous casting line, hot-worked in a plate mill, quenched to martensite, and eventually tempered to stimulate secondary precipitation and improve toughness. This production line, with a high throughput of steels, has a relatively narrow process window which puts requirements on optimal alloying to provide high cleanliness and an absence of coarser carbides. It is of interest to maximize the precipitation of secondary carbides at the tempering temperature but avoid excessive precipitation at high temperatures during hot working and austenitization.

Strong carbide-forming elements are added to the martensitic tool steels to form nano-sized secondary carbides, which precipitate at tempering, giving strength to the material. Examples of elements used as carbide formers are Ti, Nb, Mo, and V [1,2]. Steels alloyed with V and Mo were extensively studied throughout the years [3–17] following the

continuous development of new and better techniques to characterize carbides in higher resolution. Most of these studies are based on model alloys, with a limited number of alloying elements and homogenized microstructures heat-treated in controlled laboratory conditions. In this work, it is of interest to follow industrial low-alloyed tool steels, which will have a non-homogeneous microstructure due to segregations at processing.

The investigated steels are multicomponent (Fe-C-Si-Mn-Cr-Ni-Mo-V-Nb-Ti-N), which means that several types of particles can form during processing. Secondary carbides are of main interest in this work: Fe-based cementite $M_3C$, Mo-based carbides such as $M_2C$ and $M_6C$, Cr-rich carbides $M_{23}C_6$ and $M_7C_3$, and complex Mo-V mixed carbides MC. Primary carbides, carbonitrides, and nitrides are of less interest in this work.

The precipitation of carbides at quenching and tempering is often divided into four different stages [18]: (i) carbon segregation and clustering, (ii) decomposition of retained austenite, (iii) precipitation of iron carbides, and (iv) precipitation of secondary carbides [18]. The redistribution of carbon atoms is expected to take place already during quenching due to the high mobility of the carbon atom. This phenomenon is called auto-tempering and is well described in the work of Speich [19]. More recently, Hutchinson et al. [20] showed by performing APT that already after quenching, lath boundaries and dislocations were decorated with carbon atoms, which could result in the nucleation of auto-tempering carbides (AT-carbides). The nucleation and growth of AT-carbides are difficult to predict, especially for commercially produced steels, where the cooling rate is limited due to the finite thickness of the plate. Different types of iron carbides are suggested to precipitate as AT-carbides, i.e., hexagonal ε and orthorhombic η carbides [21].

Secondary carbide precipitation with metallic elements such as molybdenum and vanadium require higher temperatures for activation due to substitutional diffusion. A strengthening effect of these carbides is normally seen when tempering in the 500–600 °C range [18]. The precipitation of secondary carbides is often assumed to take place by so-called separate nucleation, with $MC/M_2C$ formation proceeding through the dissolution of cementite [16]. At longer tempering, carbides can also form by in situ transformation, with a more gradual transformation of a new carbide at the carbide/matrix interface. These secondary carbides are typically either a hexagonal $M_2C$ type or cubic MC-type of carbides. Both these carbides are solid solution phases with quite similar stability and composition. It is shown that minor differences in alloying could alter the stability of these carbides. VC-type carbides are expected to form in steels with V/Mo-ratios well below 1 since vanadium is a stronger carbide former than molybdenum. The absence of $Mo_2C$-type carbides was reported in steels with V/Mo-ratios as low as 0.23 [12]. In a study by Baker and Nutting [5], steel with a V/Mo-ratio of 0.26 was investigated, and no $Mo_2C$ carbides were detected after tempering at 400–750 °C and up to 1000 h. In the same study, it was shown that Mo could replace V in VC-type carbides with 10–50% depending on temperature [5]. Similar results were reported by Pickering [6], who studied steels with V/Mo ratios between 0.22–1.56. Two steel grades with even lower V/Mo-ratios, 0.11 and 0.17, were studied in this work.

Secondary carbides in low-alloyed steels are often small and in low phase fractions. Therefore, it is necessary to analyze large sample volumes with high-resolution techniques to resolve these precipitates and to obtain good statistics. We apply small-angle neutron scattering (SANS) in the present work to quantify the nanoscale precipitates in the bulk of the steel. Scanning transmission electron microscopy (STEM) and atom probe tomography (APT) complement the SANS measurements to provide information about the structure at the nanoscale. The microscale is covered by scanning electron microscopy (SEM), and hardness measurements provide mechanical property data. An accurate understanding of precipitation in these materials and quantitative data on precipitation is essential for the further optimization of these types of steels and to improve the physical-based modeling of the precipitation reaction [22,23]. The SANS results help to further understand the secondary hardening effect which is controlled by the number density and volume fraction of small carbides, information that is difficult to obtain with conventional methods.

## 2. Materials and Methods

### 2.1. Materials

The chemical compositions of the studied industrial low-alloyed steels are given in Table 1. Steel sheets, about 70 mm thick, which had undergone hot-working, austenitization, and oil quenching at SSAB Special Steels, were tempered at 550 °C and 600 °C up to 24 h with intermediate steps of 0, 0.5, 1, and 4 h. All tempered steel sheets were subsequently quenched in water. The sample denoted 0 h represents the ramping up to tempering temperature and direct quenching in water. It took about 1 h to reach the tempering temperature. One as-quenched sample for each material was analyzed as a reference and is denoted as AsQ. A Carbolite-Gero CWF 12/65 furnace (Carbolite-Gero, Hope, Derbyshire, England) with a 301 PID furnace controller was used for the tempering trials.

**Table 1.** Chemical composition of studied steels, values in wt.%.

| Element | Fe | C | Si | Mn | Cr | Ni | Mo | V | Ti, Nb |
|---|---|---|---|---|---|---|---|---|---|
| Material A | Bal. | 0.3 | 1.1 | 0.81 | 1.5 | 1.4 | 1.1 | 0.13 | Traces |
| Material B | Bal. | 0.3 | 1.1 | 0.81 | 1.4 | 0.7 | 0.8 | 0.14 | Traces |

### 2.2. Hardness Measurements and Optical Microscopy

Hardness measurements were performed to investigate the difference in hardness depending on alloy, tempering temperature, and time. A Qness, Q10A+, hardness tester (QATM, Golling, Salzburg, Austria) was used, and 25 measurements at HV5 were used to evaluate the hardness of each sample. Samples for optical microscopy (OM) were prepared by standard metallographic preparation and etched in 2% Nital solution.

### 2.3. Electron Microscopy

SEM imaging and energy-dispersive X-ray spectroscopy (EDX) were performed on samples that was ground and polished with SiC-paper down to 4000 grit, thereafter polished with diamond suspension, and finally polished using a $SiO_2$ suspension. The samples were gently electro-etched in a solution with 90% acetic acid and 10% perchloric acid at room temperature, 20 V for 20 s, to enhance the contrast between the matrix and the carbides.

The SEM imaging was performed in both backscattered electron and in-lens secondary electron mode in a Jeol JSM-7001F SEM (Jeol, Akishima, Tokyo, Japan). An Oxford Instruments X-MAX 80 $mm^2$ detector (Oxford Instruments, Abingdon, Oxfordshire, England) was used for the EDX-analysis. The imaging and EDX-analysis were performed with an acceleration voltage of 5 and 10 kV and a working distance of 4.5 and 10 mm, respectively.

STEM imaging and EDX were conducted on carbon extraction replicas for the detection of the metallic elements in carbides. For the preparation of carbon extraction replicas, samples were ground and polished down to 0.25 μm with diamond paste, followed by rapid etching for 5 s in 2% Nital. Afterward, a carbon film was deposited on the etched surface using a Precision Etching and Coating System (PECS) (Gatan, Pleasanton, CA, USA) to a thickness of about 20 nm. The removal of the film was conducted by immersing the samples in 10% Nital etchant until the carbon film was released from the surface. Small pieces of the carbon replica were then placed on TEM copper grids. EDX data regarding the at.% of iron, chromium, manganese, molybdenum, and vanadium were collected for each particle analyzed. Carbon and all elements below 1 at.% were excluded from the analysis.

### 2.4. Atom Probe Tomography

APT samples were cut into rods approximately 0.3·0.3·15 $mm^3$ in size using a slow-speed diamond saw. Ultra-sharp samples were then prepared using a two-stage electropolishing procedure. In the first stage, a neck was created by polishing in a layer of a strong electrolyte (10% perchloric acid, 20% glycerol, and 70% methanol) atop a heavy inert liquid (Galden). In the second stage, the entire rod was polished in a weak electrolyte (2% perchloric acid in 2-butoxyethanol) until the part below the neck was separated at

the position of the neck made in the first stage. The samples were further polished using pulsed polishing (typically 3 pulses of 3 ms duration) in the weak electrolyte to clean the surface. All polishing was conducted at around 17 V, and Pt was used as a cathode.

The atom probe used was a LEAP 3000X HR (Imago Scientific Instruments, Madison, WI, USA). The analyses were made in laser pulse mode, with a laser wavelength of 532 nm at a pulse frequency of 200 kHz, laser pulse energy 0.30 nJ, sample temperature 50 K, and evaporation rate 0.5%. The reconstructions were made using the voltage reconstruction scheme, with an image compression factor of 1.65, a k-value of 4.0, and an evaporation field of 25 V/nm. The results are presented using atom density maps. The chemical composition of particles was obtained using iso-surfaces. The iso-surfaces were obtained using a cubic voxel size of 1.0 $nm^3$ and with a delocalization of $3.0 \times 3.0 \times 1.5$ $nm^3$. APT analyses were carried out on samples from Material A in conditions: AsQ and tempered at 600 °C for 0, 4, and 24 h.

### 2.5. Small-Angle Neutron Scattering Experiment

Ex situ SANS experiments using unpolarized neutrons were performed at the Swiss spallation neutron source SINQ and the SANS-I instrument [24], Paul Scherrer Institute (PSI, Villigen, Brugg, Switzerland). Experiments were conducted using a beam of cold neutrons monochromated by a velocity selector. The wavelength ($\lambda$) of the incident neutrons was 6 Å, and the wavelength resolution ($\Delta\lambda/\lambda$) was 10%. A magnetic field of 1.5 T was applied perpendicular to the incident beam to separate the magnetic and nuclear scattering and to minimize the magnetic scattering parallel to the magnetic field. The magnetic field was assumed to be high enough to saturate the sample. The experiment was conducted at three different sample-to-detector distances, 2, 6, and 18 m. A circular aperture with a diameter of 13 mm was used to limit the incident neutron beam on the sample. The sample thickness was close to 1 mm for all samples irradiated. Calibration to absolute scale was conducted by measuring a standard water sample with a known scattering cross-section. The investigated q-range was 0.03–2.5 $nm^{-1}$, corresponding to particle diameters of about 1.3–100 nm in real space. All measurements were corrected regarding background, sample thickness, and sample transmission. A schematic of the performed SANS experiment is shown in Figure 1. The BerSANS software program (August 2014, Uwe Keiderling, Berlin, Germany) [25] was used for data correction, and the Sasfit software program (0.94.11, PSI, Villigen, Switzerland) was used for model-dependent fitting [26].

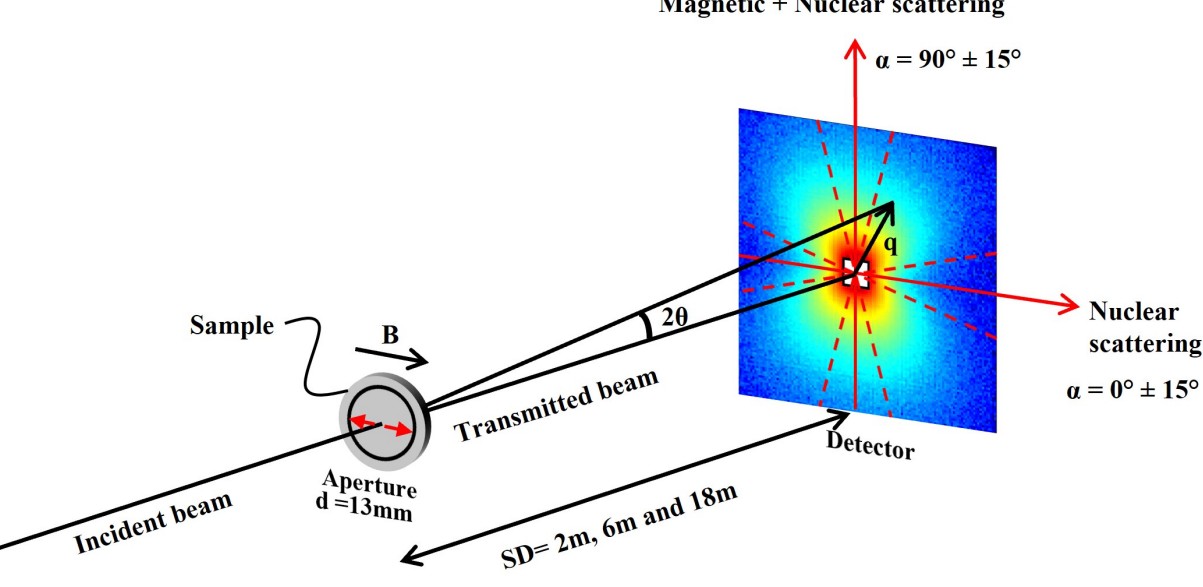

**Figure 1.** Schematic of the small-angle neutron scattering (SANS) setup used at SINQ, and the SANS-I instrument, Paul Scherrer Institute (PSI).

2.5.1. SANS Theory and Data Analysis

The following section gives a brief description of the SANS theory and data analysis procedure applied in the present work. Further details on the relevant SANS theory can be found in the work of Guinier and Fournet [27] as well as Squires [28].

The intensity $I(q)$ of neutrons coherently elastically scattered at small angles can be formulated as:

$$I(q) = (\rho_p - \rho_m)^2 N_p \left| \int_{V_p} \rho(r) e^{iqr} d^3r \right|^2 \tag{1}$$

We assume that all particles are identical with uniform scattering length density $\rho_p$ in a matrix with a uniform scattering length density $\rho_m$. If all particles are identical and their position is uncorrelated, the integral can be divided into an integral over the particle volume $V_p$ and a contrast factor, which depends on the difference between the scattering length density of the particles and the matrix [29]. By assuming that the position of the precipitates is uncorrelated, we do not consider a structure factor, i.e., we assume $S(q) = 1$. The integral describes the scattering intensity of particles randomly positioned in the matrix and has been solved analytically for different particle shapes. The differential scattering cross-section of a polydisperse and dilute system can thus be calculated by [30]:

$$I(q) = \Delta\rho^2 \int_0^\infty V(R)^2 N(R) F(q, R)^2 dR \tag{2}$$

$R$ is the radius of any particle, $\Delta\rho = (\rho_p - \rho_m)$ is the difference in the scattering length density between particle and matrix, called the scattering contrast factor, $V(R)$ is the volume of particle, $N(R)$ is the number of scattering bodies per volume with radius $R$, and $F(q, R) = \int V(R)\rho(r)e^{iqr}d^3r$ is the form factor. $F(q, R)$ depends on the size and shape of the dispersions and hence varies; several form factors for typical particle shapes found in steel are readily accessible in the literature [26]. From the microstructure analysis using SEM, TEM, and APT, two different form factors were considered, small spheres and large randomly oriented rods.

If a saturation magnetic field is applied during the measurement, the scattering pattern will become anisotropic with an isotropic nuclear scattering contribution and a $\sin^2$-term for the magnetic scattering contribution, i.e.,

$$\Delta\rho^2 = \Delta\rho_{nuc}^2 + \Delta\rho_{mag}^2 \sin\alpha^2 \tag{3}$$

with $\alpha$ being the angle between the scattering vector and the magnetic field.

In this study, sectors $0° \pm 15°$ and $180° \pm 15°$ were evaluated to consider the nuclear scattering cross-section (sectors parallel to the magnetic field), which was used to analyze the precipitation of secondary carbides.

2.5.2. Modeling of SANS Data

A Porod background from the matrix containing large particles was assumed for all measurements together with an incoherent background at high $q$, and subtracted from the scattering curves, see Figure 2. Larger particles outside the detected length-scale give scattering signals at all $q$-values. The background scattering from interfaces between matrix and large particles can be modeled by a power law of $q^{-4}$. Scattering from large particles and the isotropic background was fitted over the whole q-range for each scattering measurement by $c_0 + c_4 q^{-4}$.

For a representative plot regarding the background subtraction for the as-quenched sample, see Figure 2a,b. Red-colored data points represent the measurement after data correction, black the background (power-law plus incoherent), and blue the data after background subtraction. Scattering contrasts used for fitting are presented in Table 2. The scattering length densities were calculated from the chemical compositions measured with

APT for Fe-rich and Mo-rich carbides. Either one or two log-normal size distributions were fitted to the background corrected SANS curves through a least-square fitting process in the Sasfit software [31]. For tempered samples, a representative plot of Material A tempered at 600 °C for 1 h is given in Figure 2c, and the corresponding fitting procedure of the subtracted 1D curve is given in Figure 2d. For all tempered samples, bimodal log-normal size distributions were fitted to the 1D curve representing cementite and Mo-rich carbides.

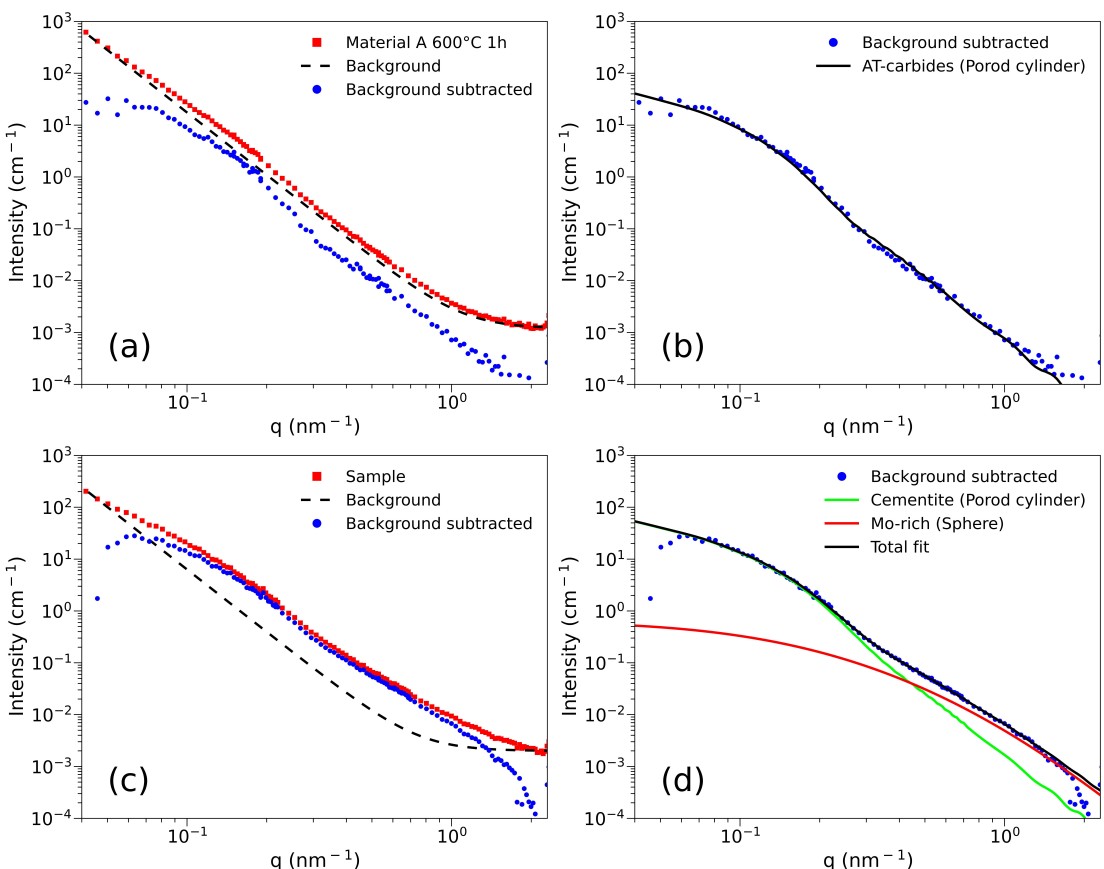

**Figure 2.** Signals without Porod background and particle fitting. The fitting procedure of Material A, AsQ is given in (**a**,**b**), and Material A 600 °C 1 h is given in (**c**,**d**).

**Table 2.** Input parameters for model-dependent fitting.

| Scatterer | Shape | SLD ($A^{-2}$) | Contrast Factor ($A^{-2}$) |
|-----------|-------|----------------|-----------------------------|
| Matrix | - | $7.8 \times 10^{-6}$ | - |
| Fe-rich | Cylindrical, *l* = 100 nm, *r* = modeled | $6.77 \times 10^{-6}$ | $1.03 \times 10^{-6}$ |
| Mo-rich | Spherical, *r* = modeled | $3.88 \times 10^{-6}$ | $3.92 \times 10^{-6}$ |

## 3. Results and Discussion

### 3.1. Hardness Measurements

Hardness (HV5) for both materials plotted against tempering time at 550 °C and 600 °C is presented in Figure 3. It can be seen, for both tempering temperatures and both materials, that the hardness decreased drastically from the AsQ-state to 0 h. Hardness stabilizes for samples tempered at 550 °C up to 4 h, after which the hardness slightly decreases up to 24 h. At 600 °C, hardness decreases continuously with time. However, between 0 h and 4 h, there is a tendency of stabilization in hardness also for samples tempered at 600 °C. Material A, which has higher alloying (Mo and Ni), had a higher hardness compared to Material B.

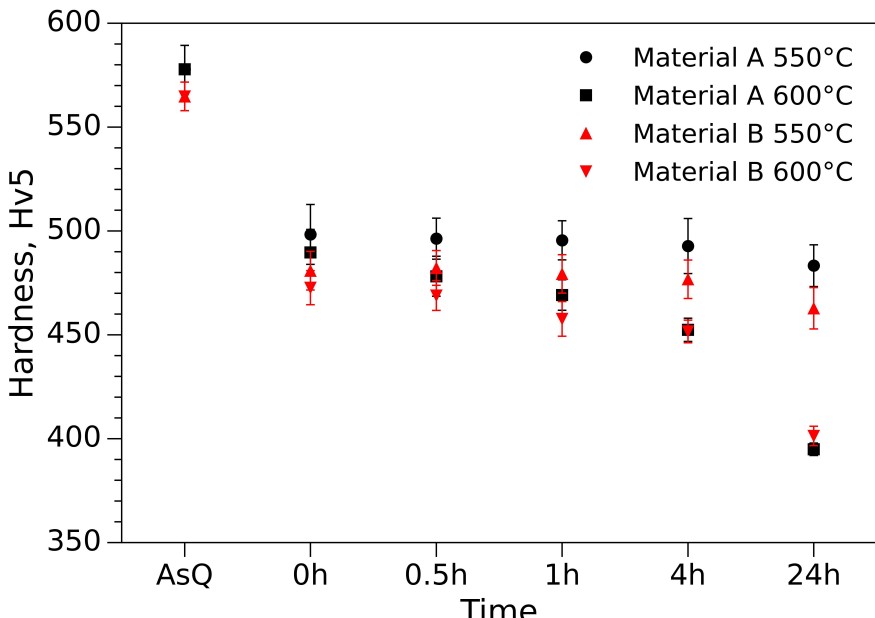

**Figure 3.** Microhardness as a function of tempering time.

### 3.2. General Microstructure

The microstructure of the as-quenched sample consisted of martensite and small contents of both bainite and retained austenite. From investigations by OM and SEM, it was evident that the microstructure underwent some segregation during steel manufacturing. For the segregation results in areas with different distributions of primary particles, see Figure 4a,b. Figure 4b show a micrograph with areas both rich in and relatively free from primary precipitates. The optical micrographs show a martensitic structure with large primary particles believed to be $M_6C$ (encircled in Figure 4b). Electro-etched samples were analyzed in SEM. The large $M_6C$ carbides were seen at prior austenite grain boundaries, see Figure 4c. This segregation of substitutional elements and precipitation of $M_6C$ carbides is assumed to locally influence the V/Mo ratio and the stability of secondary carbides.

Thin elongated AT-carbides precipitated in the martensitic blocks are shown in Figure 4c,d. Some blocks are free from AT-carbides. This could be related to inhomogeneity in chemical composition or different formation temperatures of laths during quenching giving rise to carbon diffusion to AT-carbides. The coarser particles formed at high-temperature processing were assumed to be outside the detected length scale in the SANS experiment; therefore, they would only affect the overall appearance of the SANS curve in the form of background. However, the thickness of the AT-carbides was within the detected length scale.

Representative micrographs of samples tempered for 0 and 24 h at 600 °C are given in Figure 5. After 0 h, most of the AT-carbides within the laths were dissolved and replaced with carbides precipitated on the lath boundaries, representing the dissolution of metastable iron carbides and the formation of cementite $Fe_3C$. The size of the $Fe_3C$ carbides increased with tempering time. Moreover, parallel to the formation of $Fe_3C$ carbides, indications of small particles within the martensitic laths were seen already after 0 h, see Figure 5a. However, further investigation using STEM or APT was necessary to confirm their presence. Retained austenite was seen after 0 h but was expected to be fully transformed after 24 h.

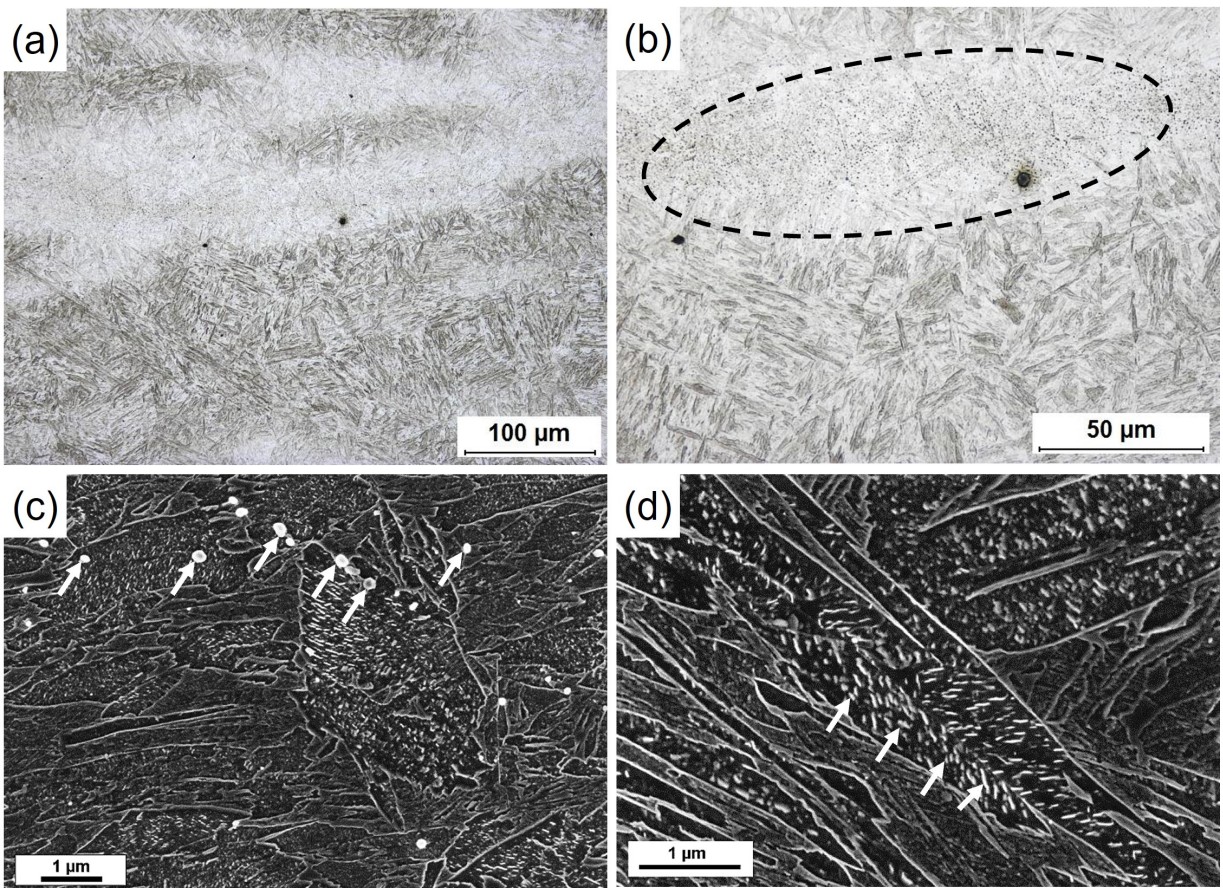

**Figure 4.** Micrographs of the AsQ-sample of Material A. (**a**) Overview of AsQ sample taken in optical microscope (OM) showing a segregated microstructure. (**b**) Micrograph showing areas rich (encircled) and poor in primary particles. (**c**,**d**) Images were taken in scanning electron microscope (SEM) on electropolished surfaces. In (**c**), arrows point at large spherical $M_6C$ carbides in lath and grain boundaries. In (**d**), arrows point at small rod-like iron carbides seen within the martensitic laths.

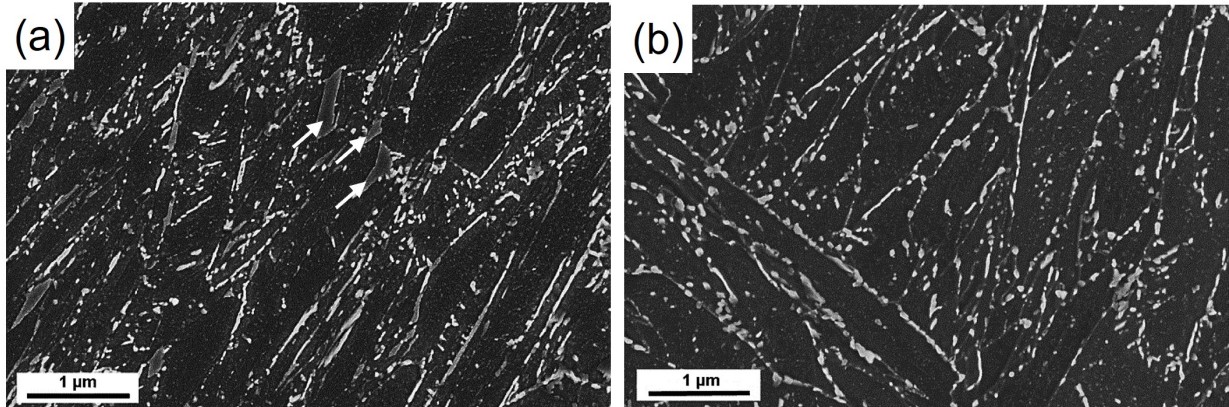

**Figure 5.** Images were taken in SEM on electropolished surfaces. Arrows point at areas of retained austenite. Rod-like particles within or on the martensitic lath boundaries are iron-carbides (**a**) Samples tempered at 600 °C for 0 h and (**b**) after 24 h.

### 3.3. Carbide Analysis with STEM

The identification of phases in STEM was conducted by studying the metal composition with EDS. Note that both the martensitic matrix and retained austenite were expected

to dissolve during the etching step during the sample preparation, and only the carbide phases were considered during STEM-EDS analysis.

STEM analysis on carbon replicas shows similar trends for materials A and B regarding the particle types and their evolution at tempering. In the AsQ-sample, relatively small Nb-rich carbides about 10 nm in diameter were detected, which are believed to precipitate during hot rolling. Some V, Mo, and Ti were found in these particles as well. A micrograph showing a representative appearance of the Nb-rich carbide is seen in Figure 6a.

AT-carbides within the martensitic laths were found on the carbon replicas from the AsQ-sample. It is believed that some AT-carbides are lost in the preparation when etched in Nital due to the high iron content and their coherency with the matrix, as discussed in the early work by Baker and Nutting [16]. However, in some martensitic laths, AT-carbides could be found. Only iron could be detected in these carbides. No secondary carbides were found in the AsQ-sample, which indicates that the quenching from austenitization was fast enough to prevent precipitation of Mo/V type secondary carbides.

The 0 h reheated sample showed cementite carbides on the carbon replica, which appeared as both spheres and elongated rod-like particles; both large and relatively small cementite carbides were seen. In contrast to the At-carbides in the AsQ state, the cementite is enriched in chromium, manganese, and molybdenum. Except for cementite carbides, both primary Nb-rich and small Mo-rich carbides were detected. The Mo-rich carbides also contain Fe, Cr, and V and can be approximated to have a spherical shape. Both cementite and Mo-rich particles are seen in the micrographs in Figure 6b–d, representing samples tempered at 0 h, 1 h, and 24 h.

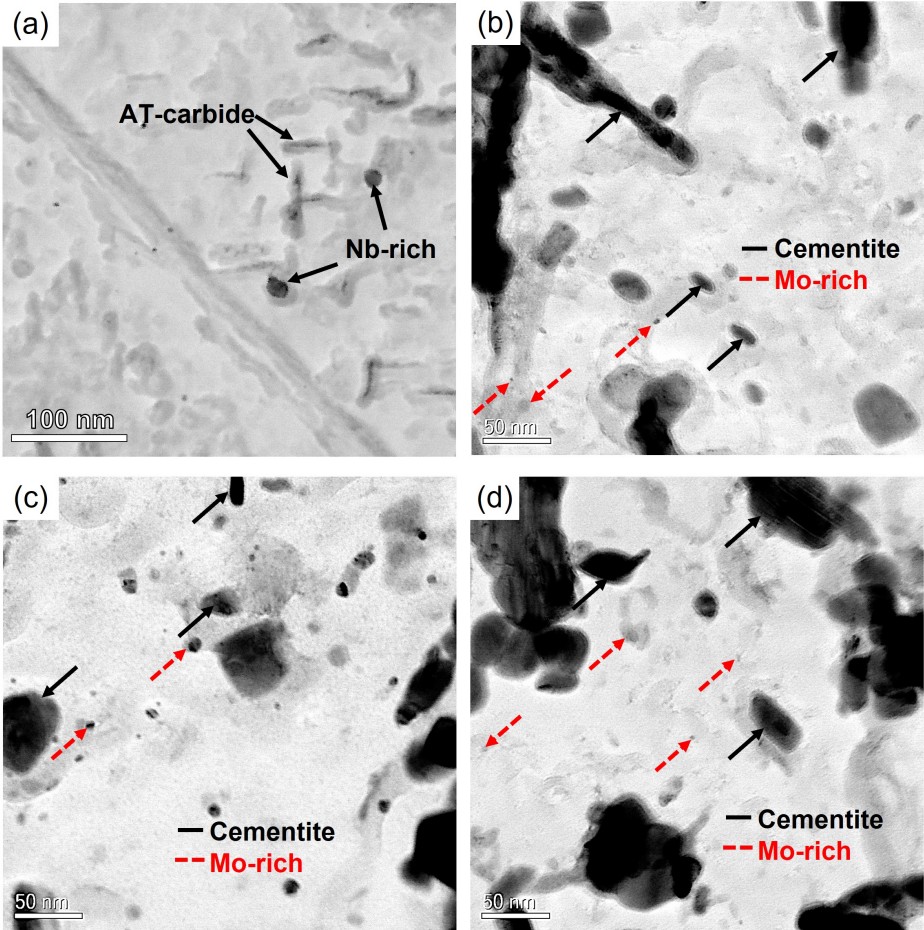

**Figure 6.** Representative bright field images from the scanning transmission electron microscopy (STEM) and energy dispersive X-ray spectroscopy EDS analysis on carbon replicas of Material A. (**a**) AsQ and samples tempered at 600 °C for (**b**) 0 h, (**c**) 1 h and (**d**) 24 h.

High Mo and low V content in the secondary carbides would indicate that they are $M_2C$. However, diffraction studies show that they are predominately MC in the early stages of tempering at 600 °C. With longer times and higher tempering temperature, the amount of $M_2C$ increases significantly, and only small amounts of $M_2C$ were found in samples tempered at 550 °C. The average metal contents are reported in Tables 3 and 4 for cementite and Mo-rich carbides, respectively. The general trend for cementite is that the Cr, Mn, and Mo increase with tempering time. For the Mo-rich particles, Mo and V increase with tempering.

**Table 3.** At% of metal constituents in cementite carbides.

| Material | °C | Method | Time, h | Fe | Cr | Mn | Mo | V |
|---|---|---|---|---|---|---|---|---|
| A | 550 | EDS | 0 | 60.1 ± 8.7 | 21.8 ± 6.4 | 2.0 ± 1.5 | 5.0 ± 2.9 | 1.4 ± 1.3 |
| | | EDS | 1 | 71.5 ± 6.9 | 15.7 ± 3.9 | 3.4 ± 1.9 | 7.0 ± 6.5 | 0.5 ± 1 |
| | | EDS | 24 | 73.0 ± 2.3 | 15.1 ± 2.9 | 4.1 ± 1.3 | 4.6 ± 1.6 | 0.6 ± 0.7 |
| | 600 | EDS | 0 | 65.1 ± 8.5 | 15.1 ± 3.2 | 2.7 ± 1.1 | 12.4 ± 8.1 | 1.7 ± 2.6 |
| | | EDS | 1 | 68.3 ± 7.3 | 17.5 ± 5.2 | 5.5 ± 1.1 | 5.7 ± 9.0 | 1.0 ± 4.9 |
| | | EDS | 24 | 68.4 ± 6.7 | 18.3 ± 5.8 | 6.8 ± 0.8 | 4.6 ± 1.6 | 1.1 ± 0.7 |
| B | 550 | EDS | 0 | 78.4 ± 5.4 | 14 ± 4.3 | 3.1 ± 2.0 | 2.5 ± 0.9 | - |
| | | EDS | 1 | 74.9 ± 4.5 | 14.6 ± 4.0 | 3.5 ± 2.1 | 3.9 ± 1.2 | - |
| | | EDS | 24 | 63.5 ± 7.8 | 19.2 ± 5.5 | 4.5 ± 2.2 | 8.0 ± 6.1 | 1.7 ± 0.9 |
| | 600 | EDS | 0 | 79.0 ± 5.2 | 12.2 ± 3.1 | 2.9 ± 1.5 | 4.5 ± 2.1 | - |
| | | EDS | 24 | 69.7 ± 4.8 | 17.5 ± 3.7 | 5.2 ± 0.9 | 6.0 ± 1.6 | - |

**Table 4.** At% of metal constituents in Mo-rich carbides.

| Material | °C | Method | Time, h | Fe | Cr | Mn | Mo | V |
|---|---|---|---|---|---|---|---|---|
| A | 550 | EDS | 0 | 43.1 ± 4.4 | 22.2 ± 5.7 | - | 26.7 ± 3.3 | 2.5 ± 1.8 |
| | | EDS | 24 | 46.0 ± 2.5 | 29.1 ± 6.8 | - | 13.8 ± 5.8 | 1.7 ± 1.0 |
| | 600 | EDS | 0 | 24.9 ± 8.5 | 13.0 ± 3.2 | - | 51.9 ± 8.1 | 6.5 ± 2.6 |
| | | EDS | 1 | 20.4 ± 7.3 | 11.1 ± 5.2 | - | 60.9 ± 9.0 | 5.5 ± 4.9 |
| | | EDS | 24 | 13.7 ± 4.8 | 12.3 ± 4.9 | - | 61.0 ± 3.6 | 11.4 ± 3.5 |
| B | 550 | EDS | 0 | 44.1 ± 2.4 | 30.9 ± 3.4 | - | 16.1 ± 3.8 | - |
| | | EDS | 1 | 44.0 ± 2.8 | 29.3 ± 5.0 | - | 14.0 ± 1.6 | 1.7 ± 1.3 |
| | | EDS | 24 | 36.4 ± 8.1 | 17.5 ± 7.1 | - | 32.2 ± 3.7 | 5.9 ± 5.0 |
| | 600 | EDS | 0 | 29.9 ± 3.7 | 14.7 ± 2.3 | - | 45.2 ± 3.8 | 7.6 ± 4.0 |
| | | EDS | 24 | 12.4 ± 4.3 | 14.7 ± 3.4 | - | 60.1 ± 2.5 | 10.5 ± 3.9 |

*3.4. Carbide Analysis with APT*

APT measurements were performed on four different samples from Material A, AsQ, and samples tempered at 600 °C for 0, 4, and 24 h. In the AsQ sample, only C was not randomly distributed. C was found to be segregated to martensitic lath boundaries, although no distinct particles were detected in this analyzed volume, see Figure 7a.

For the material reheated to tempering temperature, 0 h sample, the rearrangement of metallic elements occurred. Large, elongated cementite carbides were seen, including Cr and Mn and a minor concentration of Mo and V. A second population of carbides was seen, rich in Mo and V, located both at dislocations and isolated within the martensitic laths. Both Cr and Mn were relatively evenly distributed in the matrix and were maintained at longer tempering times. The content of Mo, and especially V, decreased in the matrix with tempering.

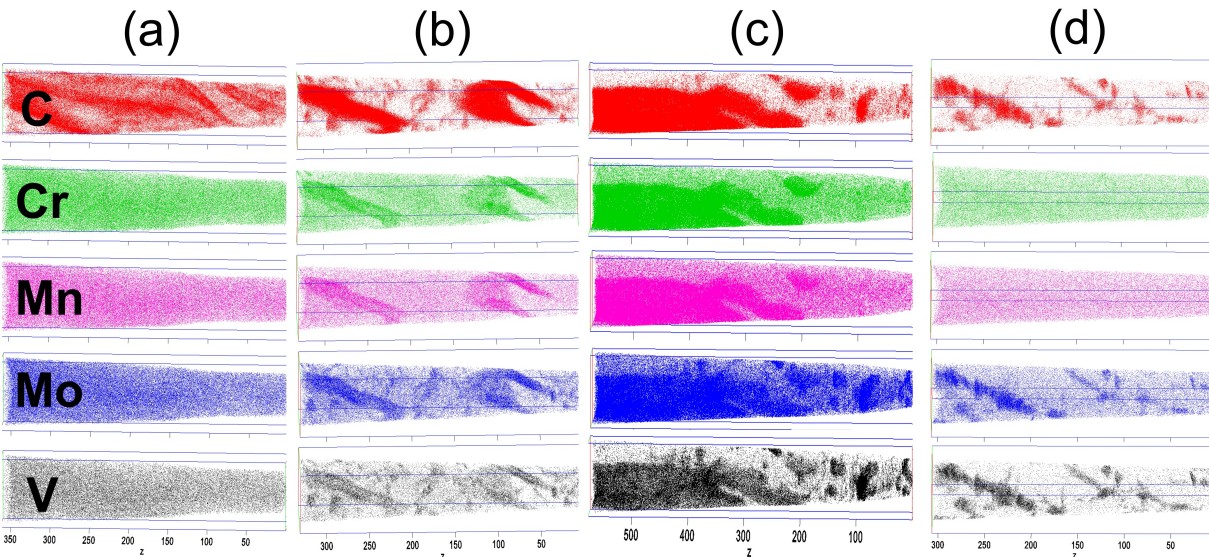

**Figure 7.** APT results for material A tempered at 600 °C. Distribution of C, Cr, Mn, Mo, and V after (**a**) quenching, (**b**) 0 h at 600 °C, (**c**) 4 h at 600 °C, and (**d**) 24 h at 600 °C (axes in nm).

To approximate the size of formed carbides, iso-surfaces were created with specific boundary thresholds regarding the limiting element composition within each particle type. From each particle type, the average chemical composition was calculated for all particles defined as cementite or Mo-rich. For the 0 h sample, iso-surfaces of Mo-rich carbides were created with Mo + V > 10 at.%. The concentration of Fe in the Mo-rich carbides was relatively high; however, the ion density was low, which is a known artifact of APT analysis for alloy carbides such as $(Mo, V)_2C$ and $(Mo, V)C$ [32], a consequence of the high evaporation field of such carbides resulting in local magnification [33]. The diameter of isolated and rounded Mo-rich carbides is typically 6 nm, whereas the diameter of the elongated Mo-rich carbides on dislocations is about 2 nm. There is no difference in composition when comparing the rounded and elongated carbides. It should be noted that most carbides appear larger in the APT reconstruction due to the local magnification effect. In the current analyses, the carbide volume was estimated to be about three times larger than the true volume, equivalent to the 1D magnification being 1.7 times too large, from the observed density variations [32,33]. This effect does not influence cementite.

After 4 h of tempering, more distinct Mo-rich carbides were formed; see element distribution in Figure 7c. The iso-surface threshold for Mo-rich carbides was set to Mo + V > 30 at.%. Compared to the 0 h sample, the Mo-rich carbides were larger. The rounded carbides had a diameter of typically 9 nm, and the carbides at dislocations had a diameter of about 2 nm. Large, elongated cementite carbides were also detected in the 4 h sample.

The radii of the analyzed cementite particles in the 0 and 4 h samples were about 15–30 nm, and the length was 100–200 nm (although they can be larger, as the analyzed volumes are of this order). The radius increased with tempering time. At 24 h, no cementite carbides were detected, see Figure 7d. However, both large and small cementite particles are known to be present in the STEM results.

For the 24 h sample, the iso-surface threshold for Mo-rich carbides was set to Mo + V > 30 at.%. Furthermore, two types of Mo-rich carbides were encountered in the 24 h sample located in martensitic laths, small carbides such as Mo-rich carbides found in the 0 h and 4 h samples, and one large carbide in the same APT reconstruction with higher Mo content and lower C content which would indicate an $M_2C$ structure. This indicates that the Mo-rich carbides are MC in the early stages of tempering and that the amount of $M_2C$ is increasing with tempering time at 600 °C. Whether this transformation can be characterized as separate nucleation or as in situ could not be determined. The average size of the Mo-rich

carbides was close to 8 nm after 24 h, about the same as after 4 h. The average composition of Mo-rich carbides from the APT analysis is presented in Table 5.

**Table 5.** Chemical composition (at%) of Mo-rich carbides found in samples tempered at 600 °C.

| Time, h | Type | C | Si | Mn | Cr | Mo | V | Fe |
|---------|------|---|----|----|----|----|---|----|
| 0 | MC | $15.4 \pm 0.1$ | $2.80 \pm 0.1$ | $1.40 \pm 0.1$ | $9.1 \pm 0.2$ | $12.9 \pm 0.2$ | $3.7 \pm 0.1$ | $53.4 \pm 0.3$ |
| 4 | MC | $37.8 \pm 0.3$ | $0.48 \pm 0.1$ | $0.53 \pm 0.1$ | $6.0 \pm 0.2$ | $34.1 \pm 0.4$ | $11.8 \pm 0.2$ | $8.5 \pm 0.3$ |
| 24 | MC | $34.0 \pm 0.2$ | $0.64 \pm 0.1$ | $0.58 \pm 0.1$ | $8.4 \pm 0.2$ | $33.0 \pm 0.3$ | $11.1 \pm 0.2$ | $11.6 \pm 0.2$ |
| 24 | $M_2C$ | $30.7 \pm 0.6$ | $0.20 \pm 0.1$ | $0.97 \pm 0.3$ | $9.0 \pm 0.5$ | $51.1 \pm 0.8$ | $4.9 \pm 0.3$ | $2.5 \pm 0.3$ |

*3.5. Carbide Analysis with SANS*

Plots showing averaged SANS data of Materials A and B are given in Figure 8a–d, for tempering at both 550 and 600 °C. At 550 °C, there is a steady increase in intensity at high *q*-values from the AsQ state up to 24 h. For 600 °C tempering, a similar effect is seen when comparing the AsQ condition to 0 h with an increase at high *q*-values. However, in contrast to the results at 550 °C, the intensity is relatively stable from 0 h up to 4 h at high-*q*. With further tempering at 600 °C, the intensity at high *q*-values then decreases at 24 h.

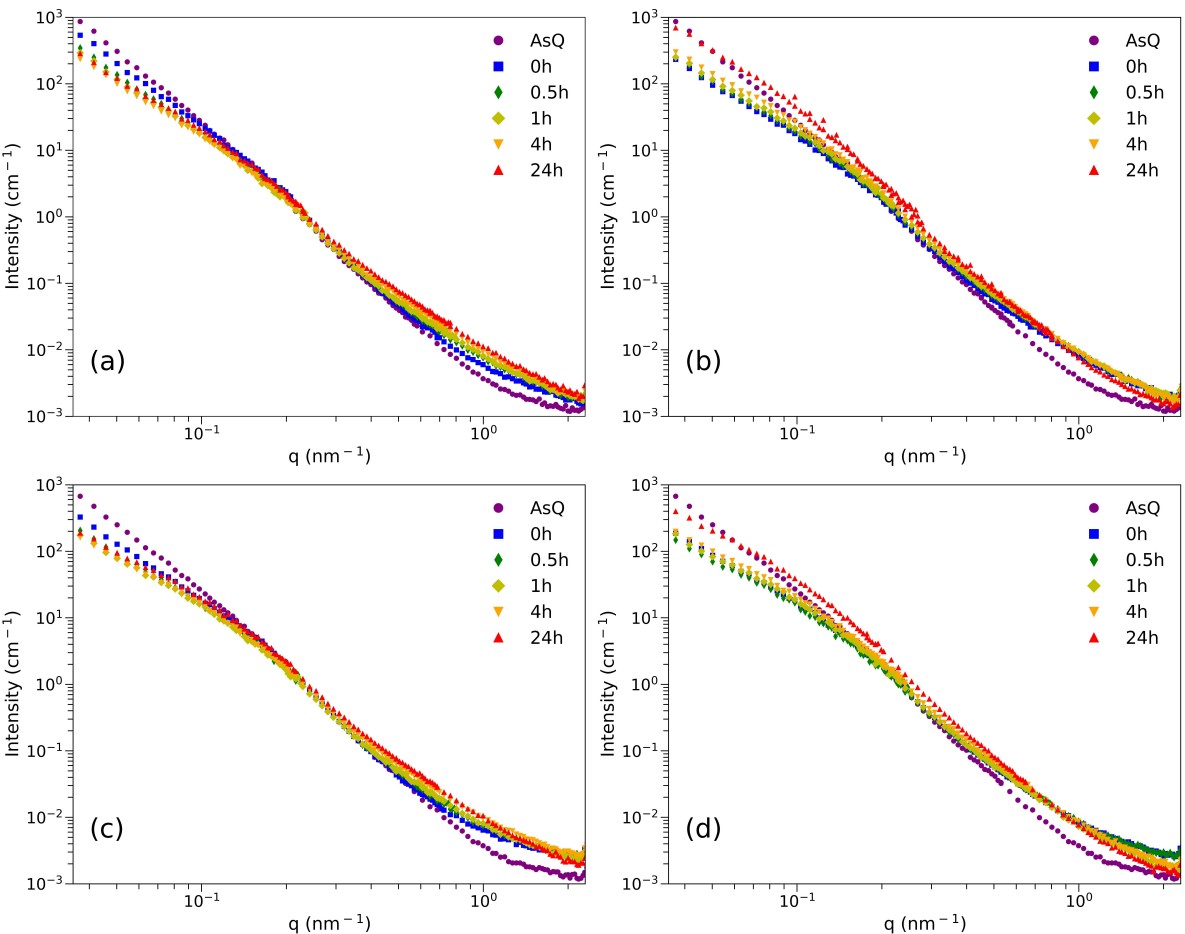

**Figure 8.** Measured SANS data of tempered samples. (**a**) Material A at 550 °C, (**b**) Material A 600 °C, (**c**) Material B at 550 °C, and (**d**) Material B at 600 °C.

For samples tempered at 550 °C, there is a continuous drop in intensity at low *q*-values from AsQ to 0 h and 0.5 h, after which the intensity is relatively stable up to 24 h of tempering. This could be the dissolution of AT-carbides formed during quenching, followed by the rapid formation and growth of cementite on lath boundaries in the same *q*-range

during tempering. This is supported by the microscopy examination. Similar trends regarding the dissolution of AT-carbides were reported by Leitner et al. [34].

For 600 °C tempering, a drop in intensity from AsQ-state to 0 h is also seen at low $q$-values. In contrast to the results at 550 °C, the intensity then increases from 0 to 24 h, which can be related to the growth of relatively large cementite carbides. Similar trends can be seen for Material B at both tempering temperatures.

Background subtraction and model-dependent fitting were performed as described in Section 2.5.2. From the microscopy investigations, it is known that Nb-rich carbides and AT-carbides are present in the AsQ-sample and within the detected length scale. A weak knee can be seen in the scattering data, related to relatively large particles with a wide size distribution. Niobium carbides are considered small (from TEM, about 10 nm in size) and few and they are assumed to disappear in the signal from other features. It is therefore believed that the signal in the AsQ-samples is from AT-carbides formed during quenching. From the microscopy investigations, they should give a signal in the current q-range (at low $q$-values).

Size distributions normalized to particle volume are presented in Figure 9. The difference between AsQ and 0 h at low q, as mentioned above, is believed to be related to the dissolution of AT-carbides and the formation of Cr and Mn enriched cementite at lath boundaries. The Mo-rich carbides are in the 1–10 nm size range, which is finer compared to cementite which is 5–30 nm in size. Comparing the effect of tempering at 550 and 600 °C, lower temperature yields more Mo-rich carbides, which are also more stable with time. For 550 °C, the number of Mo-carbides continues to increase up to 24 h, whereas at 600 °C, a peak value is reached after 1 h. The more stable particle structure at 550 °C is one reason for the more stable hardness values compared to 600 °C tempered material.

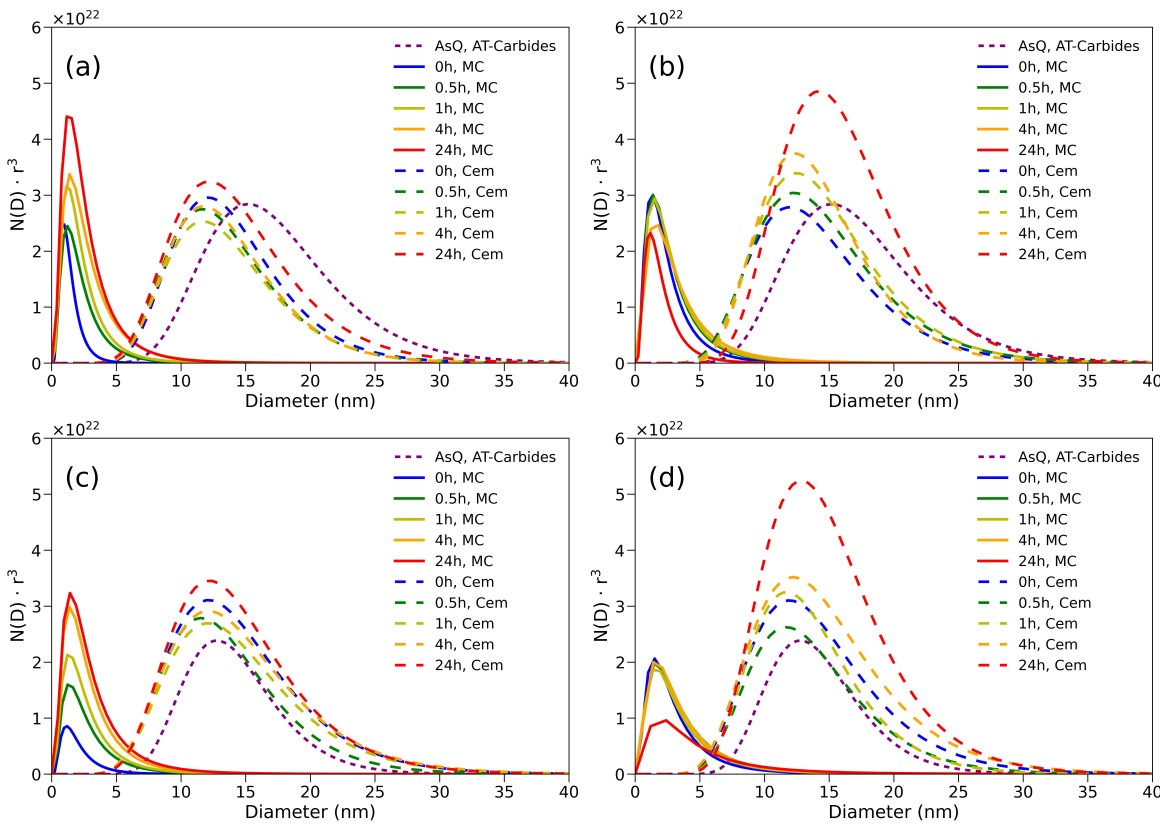

**Figure 9.** Particle volume normalized size distributions. (**a**) Material A 550 °C, (**b**) Material A 600 °C, (**c**) Material B 550 °C, (**d**) Material B 600 °C.

After 4 h tempering at 600 °C, the peak of the normalized size distribution of Mo-carbides decreases with further tempering time, and its size distribution starts to overlap

with cementite. This decrease in intensity could be a result of MC to $M_2C$ transformation, which is accompanied by an overall size increase. Similar results were reported earlier [5,35], where the enrichment of Mo is said to increase the growth rate of MC. The SANS results agree with the STEM results that showed that the cementite and Mo-rich carbides had overlapping size distributions, and the collective information from SANS, STEM, and APT provides a strong indication of the transformation from MC to $M_2C$ after 4 h at 600 °C. This effect is not seen for samples tempered at 550 °C.

The presence of carbides such as cementite and Mo-carbides in the same size range makes it difficult to determine the absolute carbide fractions with SANS. The cementite growth is restricted due to the partitioning effect of Cr, Mn, and Si, which hinder further growth. However, the high number density of Mo-rich carbides and the relatively large difference in SLD-contrast between these carbides make it possible to study the overall evolution of nano-sized carbides.

The number densities of cementite and Mo-rich carbides are presented in Figure 10a. The Mo-rich carbides are present in a much higher density compared to cementite. Their high density is believed to be an efficient obstacle for dislocation movement, thereby strengthening the material. At 600 °C, the total hardness drops continuously with time which is believed to be due to dislocation recovery and softening, which is to some extent hindered by the relatively stable Mo-carbides in the structure. When tempering up to 24 h, even the Mo-carbides grow, and the number density decreases.

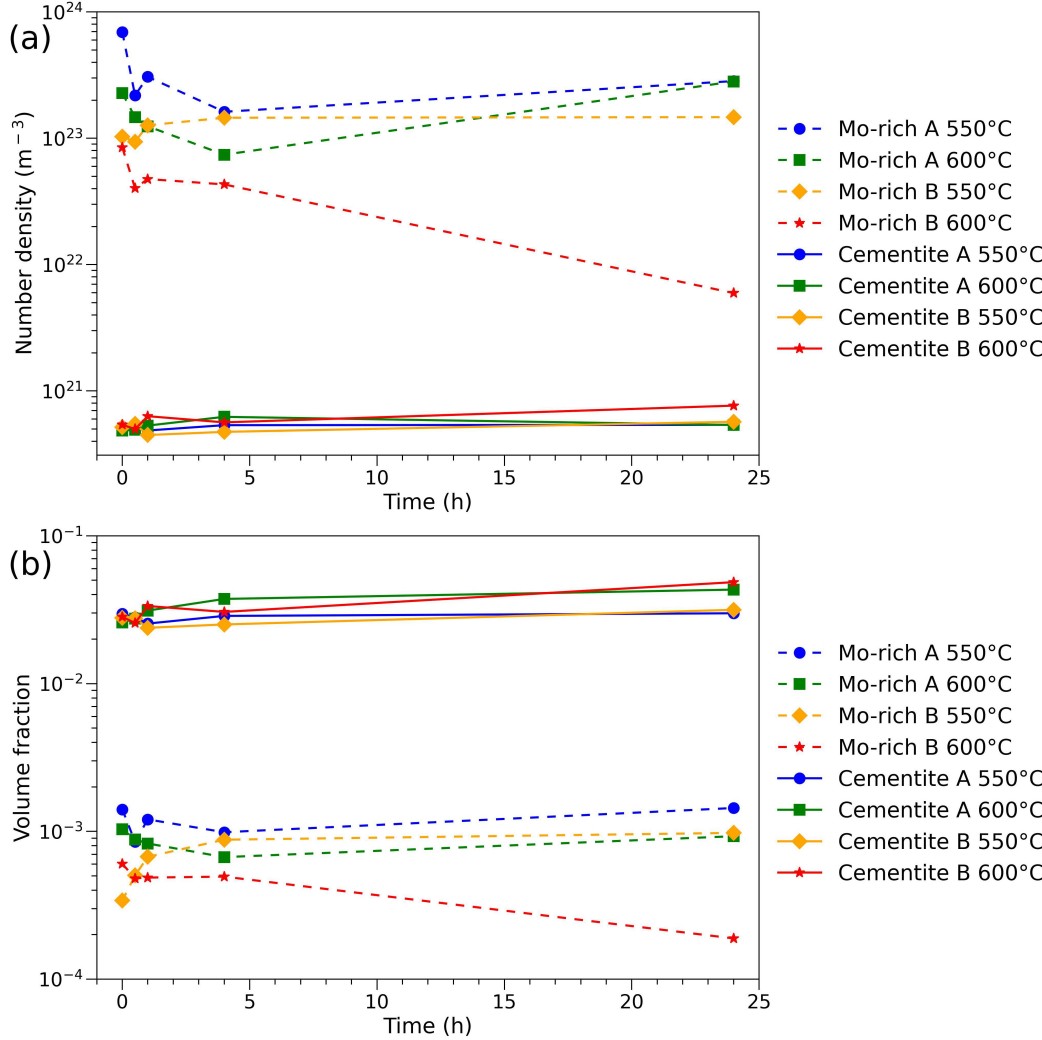

**Figure 10.** Results from the SANS modeling for both materials and tempering temperatures. (**a**) Number density and (**b**) volume fraction.

The volume fraction of cementite and Mo-rich carbides as a function of time and temperature at tempering is shown in Figure 10b. The fraction of Mo-rich carbides increases with time at 550 °C up to 24 h, whereas the 600 °C tempering reaches a maximum after 0–0.5 h. This effect is believed to be due to a strongly temperature-activated precipitation sequence of Mo-rich particles. On the other hand, the cementite shows quite a stable fraction.

## 4. Conclusions

- The high hardness in the as-quenched materials remains at longer tempering times at 550 °C compared to 600 °C. The recovery and softening of the martensitic structure are compensated by the continuous precipitation of Mo-rich carbides occurring up to 24 h at 550 °C. Due to the diffusion-activated precipitation, the nucleation and growth of Mo-rich carbides seem finished already within 1 h of tempering at 600 °C.
- Iron carbide precipitate as auto-tempering carbides at quenching, confirmed by SEM and STEM. On reheating to tempering temperatures at 550–600 °C, most of the At-carbides dissolve related to the formation of chromium and manganese enriched cementite at lath boundaries.
- At later stages of tempering, two different particle populations were seen in smaller size ranges up to 20 nm. TEM and APT identified these to be cementite (5–20 nm in radius) or Mo and V rich (1–5 nm in radius). The same results could be interpreted by the SANS results. The size distributions were determined using model-dependent fitting.
- At the early stages of tempering and after 4 h at 600 °C, TEM and SANS results show the presence of both cementite and Mo and V-rich carbides in the same size ranges.
- The estimation of both number densities and volume fractions with SANS provides additional understanding of the overall carbide evolution during tempering and its impact on mechanical properties, data that are difficult to accurately obtain with conventional methods.

**Author Contributions:** Conceptualization, E.C., P.H., M.A. and H.M.; methodology, E.C., P.H., M.A. and H.M.; validation, H.M., P.H. and E.C.; formal analysis, E.C., J.K., M.T. and F.L.; investigation, E.C.; resources, M.A., H.M. and P.H.; data curation, E.C.; writing—original draft preparation, E.C.; writing—review and editing, E.C., H.M., P.H., J.K. and M.T.; visualization, E.C.; supervision, H.M. and P.H.; project administration, H.M. and E.C.; funding acquisition, H.M. and P.H. All authors have read and agreed to the published version of the manuscript.

**Funding:** The authors acknowledge the financial support from the Swedish Foundation for Strategic Research under grant No. FID15-0043.

**Data Availability Statement:** The data presented in this study are available from the corresponding author upon reasonable request.

**Acknowledgments:** SSAB special steel is gratefully acknowledged for their financial support and for providing the experimental materials. This work is based on experiments performed at the Swiss spallation neutron source SINQ, Paul Scherrer Institute, Villigen, Switzerland.

**Conflicts of Interest:** The authors declare no conflict of interest.

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
