# Peer review of "Carbide Precipitation during Processing of Two Low-Alloyed Martensitic Tool Steels with 0.11 and 0.17 V/Mo Ratios Studied by Neutron Scattering, Electron Microscopy and Atom Probe"

_metals, doi:10.3390/met12050758_

Round 1
Reviewer 1 Report
The research report is well written and formulated. The experimental camping is complete and provides a robust support to the conclusions. Then, the article can be accepted in present form.
Author Response
The research report is well written and formulated. The experimental camping is complete and provides a robust support to the conclusions. Then, the article can be accepted in present form.
Thank you for your review
Reviewer 2 Report
This manuscript investigated the precipitation behaviors of carbides in two low-alloyed martensitic tool steels during processing by systematically advanced characterizations. It is an interesting and comprehensive job. I recommend acceptance of this manuscript after the following minor revision.
- Page 6 line 237: The first paragraph below section 3 should be deleted.
- It seems that the arrows in Figure 4 are in the wrong places. Please check and correct them.
- The figure caption of Figure 10 should be improved.
- The authors mentioned Figure 11 in line 405, but there is no Figure 11 in the manuscript. Please check.
Author Response
- Page 6 line 237: The first paragraph below section 3 should be deleted.
This paragraph has been removed
- It seems that the arrows in Figure 4 are in the wrong places. Please check and correct them.
This is corrected
- The figure caption of Figure 10 should be improved.
The figure caption has been improved with the following text: Results from the SANS modeling for both materials and tempering temperatures. a) Number density, and b) volume fraction.
- The authors mentioned Figure 11 in line 405, but there is no Figure 11 in the manuscript. Please check.
This has been corrected.
Reviewer 3 Report
This paper describes an effort to characterize two low-alloyed martensitic tool steels with neutron scattering, electron microscopy and atom probe microscopy. The paper is well written with good logical structure and fluent language. The authors combined SANS, TEM and APT characterizations, which is very nice. The results and discussions are well organized and the conclusions are supported by the experimental results. The size distributions of different types of precipitates were obtained with SANS data modeling and the evolution of precipitates are well backed up with characterization by the combined techniques.
There are occasionally minor grammar or spelling mistakes such as line 77 interphase and line 221 Fig 2 and b, etc. So the authors are recommended to check the paper throughout.
Author Response
There are occasionally minor grammar or spelling mistakes such as line 77 interphase and line 221 Fig 2 and b, etc. So the authors are recommended to check the paper throughout.
The suggested spelling mistakes has been corrected. The overall text will be checked for grammar and spelling mistakes.
Reviewer 4 Report
Overall the paper is pretty well written. The analyzed data could be useful for the research society. This paper includes data of small angle neutron scattering. This is a special technique not existing in regular lab. The advantages of SANS analyses should be addressed in the introduction section. What new aspects have been discovered with neutron scattering? This part should be emphasized in the introduction section and conclusion (summary) section.
Author Response
Overall the paper is pretty well written. The analyzed data could be useful for the research society. This paper includes data of small angle neutron scattering. This is a special technique not existing in regular lab. The advantages of SANS analyses should be addressed in the introduction section. What new aspects have been discovered with neutron scattering? This part should be emphasized in the introduction section and conclusion (summary) section.
I have added a final sentence in the introduction to capture your comment: The SANS results help to further understand the secondary hardening effect which is controlled by the number density and volume fraction of small carbides. Information that is difficult to obtain with conventional methods.
I have also added a bullet point in the conclusion: The estimation of both number densities and volume fractions with SANS provide additional understanding of the overall carbide evolution during tempering, and its impact on mechanical properties. Data which is difficult to accurately obtain with conventional methods.